# Twenty-Three Months Repetitive Transcranial Magnetic Stimulation of the Primary Motor Cortex for Refractory Trigeminal Neuralgia: A Single-Case Study

**DOI:** 10.3390/life13010126

**Published:** 2023-01-02

**Authors:** Sascha Freigang, Shane Fresnoza, Christian Lehner, Dominyka Jasinskaitė, Kariem Mahdy Ali, Karla Zaar, Michael Mokry

**Affiliations:** 1Department of Neurosurgery, Medical University Graz, 8036 Graz, Austria; 2Institute of Psychology, University of Graz, 8010 Graz, Austria; 3BioTechMed, 8010 Graz, Austria; 4Faculty of Medicine, Lithuanian University of Health Sciences Kaunas, 44307 Kaunas, Lithuania

**Keywords:** neuromodulation, chronic pain, refractory trigeminal neuralgia, rTMS

## Abstract

Treatment refractory or recurrent trigeminal neuralgia (TN) is a severe chronic pain illness. Single-session repetitive transcranial magnetic stimulation (rTMS) has been shown to elicit analgesic effects in several craniofacial pain syndromes, including TN. However, the safety and long-term effect of multi-session rTMS for TN have yet to be fully explored. In this study, we present a case of a patient with medical treatment-refractory TN after microvascular decompression. The patient volunteered to undergo 73 sessions of 10 Hz rTMS over 23 months. Neurovagination was used for precise localization and stimulation of the hand and face representation at the left motor cortex. The numeric pain intensity scores derived using the visual analog scale served as a daily index of treatment efficacy. The patient experienced a significant weekly reduction in pain scores, cumulating in 70.89% overall pain relief. The medication dosages were reduced and then discontinued toward the end of the intervention period. No severe adverse events were reported. From our results, we can conclude that the longitudinal multi-session application of rTMS over the hand and face area of M1 is a safe and effective method for producing long-lasting pain relief in TN. Using rTMS may thus prove helpful as an adjunct to conventional methods for treating pain in TN.

## 1. Introduction

Trigeminal neuralgia (TN) is a recurrent, sudden onset electric shock-like facial pain [1]. TN is generally considered to be a unilateral disease, but a certain number of cases of bilateral pain have been recorded [2,3]. The prevalence of TN increases with age and affects women (60%) more than men (40%) [4]. A detailed clinical history and physical examination provide sufficient diagnostic information; however, magnetic resonance imaging (MRI) is routinely performed to determine whether the pain is caused by trigeminal nerve abnormalities such as atrophy due to vascular compression (classical TN) or other causes, including multiple sclerosis or cerebellopontine angle tumors (secondary TN). The pathophysiology of TN is still debatable; genetic alterations, particularly gain-of-function mutations or altered expressions of voltage-gated sodium channels (NaV 1.3, 1.6, 1.7, and 1.8), are thought to play a role in triggering pain [5,6]. This may form the basis of why sodium channel blockers (e.g., carbamazepine) provide excellent pain relief, as they suppress ectopic discharges without blocking normal nerve conduction at a proper dose range [7]. Demyelinated afferents also tend to become hyperexcitable and capable of generating ectopic impulses manifesting as spontaneous pain [8]. Demyelination is one of the key biopsy findings on the trigeminal nerve compressed by blood vessels at the root entry zone. The transition of peripheral Schwann cell myelination to central oligodendroglia myelination takes place in the root entry zone, making it susceptible to pressure [4]. Indeed, effective pain relief is achieved with open fossa surgical decompression of the trigeminal nerve from conflicting blood vessels in 95% of TN cases [1,4].

The current pain management for TN seems adequate but far from optimal because it is ineffective in some patients, and relapse can occur. In addition, pharmacologic treatment has systemic side effects (e.g., sedation, hyponatremia, and allergies), while palliative destructive surgical procedures have complications (e.g., facial numbness or anesthesia dolorosa). In recent decades, several groups have shown the efficacy of novel repetitive transcranial magnetic stimulation (rTMS) in various facial pain syndromes. RTMS delivers electromagnetic pulses to influence the excitability of cortical areas involved in pain modulation. For TN, single-session rTMS stimulation of the hand area representation of the primary motor cortex (M1) yielded analgesic effects when applied bilaterally at 10 Hz [9] or after 5 daily 20 Hz stimulations of M1 contralateral to the painful side [10]. In one patient diagnosed with treatment-resistant depression and TN, 10 Hz rTMS of the left dorsolateral prefrontal cortex alleviated pain [11]. Prolonged/repeated rTMS stimulation (>1 week) to elicit long-lasting pain relief was also tested in a single refractory TN patient. She received 10 Hz rTMS to the contralateral M1 hand region over 1 year (35 sessions with various intersession intervals) and experienced significant pain relief limited to about 2–4 weeks after each stimulation period with no severe adverse effects [12]. In this study, we present a case of a patient with refractory TN who underwent 73 rTMS treatment sessions over 23 months. The aim of reporting is to demonstrate further the safety and long-term effect of repeated rTMS stimulation in TN patients.

## 2. Materials and Methods

### 2.1. Patient

The patient is a 64-year-old right-handed caucasian female recruited from the Department of Neurosurgery at the Medical University Graz. She was diagnosed with a right-sided TN secondary to neurovascular compression, as confirmed by an MRI scan taken on 2 May 2019 (Figure 1). She described having severe, piercing, sharp pain on the right side of her face for the past six months before her initial neurological consultation. The pain was episodic (lasting on average less than five minutes) and typically triggered by eating, talking, brushing her teeth, or washing and touching her face. The facial pain is not associated with a headache or aura. She had no family history of TG and denied having dental procedures or trauma to the head and face before the pain started. The patient was initially started with an oral antiepileptic BID or twice daily (Oxcarbazepine 375 mg) and analgesic medications that must be taken as needed (Dronabinol 2.5%, Metamizole 575 mg, and Duloxetine 30 mg). Nevertheless, the facial pain worsened, and she decided to undergo microvascular decompression of the right trigeminal nerve at our institution on August 2019, three years after the start of the medical treatment. There were no reported adverse events related to the procedure.

The decompression resulted in effective pain relief for six months. Unfortunately, her right-sided facial pain gradually relapsed and became refractory to medications. 3D T1 and T2-weighted MR images were obtained (Siemens Medical Systems, Erlangen, Germany), confirming the absence of new neurovascular compression. RTMS was offered as an experimental treatment option, to which she gave written informed consent. There were no absolute contraindications to TMS, such as metallic or electronic implants anywhere in the head, neck, or chest, or relative contraindications, such as neoplastic, infectious, or metabolic brain lesions and history of seizures or epilepsy. Her medical history revealed mild depressive episodes (ICD code: F32.0), but the patient did not disclose further information. She reported that she had blood pressure elevation starting around July 2019, which could be secondary to the episodes of facial pain (secondary hypertension). Nonetheless, her blood pressure was controlled during the study. The patient received the first rTMS stimulation on 9 June 2020. This single case study is included in our “pain project” approved by the Medal University of Graz Ethics Committee (registration number: 30-459-ex 17/18). However, specific approval is not required for this single case since the intervention was implemented as part of clinical care [13]. All procedures conform to the Declaration of Helsinki regarding human experimentation.

### 2.2. Experimental Assessment and Therapeutic Intervention

The course of the treatment follows an AB or phase change without reversal design [14]. “A” refers to the baseline phase or the period without intervention, and “B” to the intervention/rTMS phase. Measurements without intervention were not repeated after phase B; hence there is no reversal in the design (e.g., ABA design). The baseline pain measurements were conducted on four consecutive days without stimulation (week 1) (Figure 1). Following the baseline phase is the intervention phase consisting of 73 rTMS sessions spread over 23 months (weeks 2 to 89). For the intervention phase, we employed an adaptive or response-guided approach, meaning the number of stimulation sessions changed per week based on the patient’s response (pain intensity). The patient had 4 consecutive days of stimulation in weeks 2 and 3 (with 3-day between-week intervals). The between-session intervals were increased gradually such that the patient received rTMS stimulation 3×/week in weeks 4–6 (with 1-day intervals), 2×/week in weeks 7–11 (with 2- to 3-day intervals), and once per week in weeks 12–15 (with 6- to 7-day intervals). Subsequently, the stimulation was given to the patient once per week every other week (weeks 16–30). Because of good pain control, a 21-day break from stimulation was implemented (weeks 31–33). However, pain episodes and intensities increased after this break, particularly in week 36. The team decided to perform the stimulation session once weekly in the succeeding weeks (weeks 34–44). After that, the stimulation was conducted once per week with 1- or 2-week intervals (weeks 45–89), which was only interrupted by a 28-day break (weeks 70–73) and another 21-day break (weeks 86–88). Despite good pain control with regular rTMS stimulation, the patient underwent percutaneous balloon compression in week 81. After the procedure, pain control improved, but the patient developed numbness in the right facial area. The rTMS stimulation was continued for two more months with wider inter-session intervals (one to two weeks) and stopped because the patient was pain-free.

Oxcarbazepine 375 mg BID was maintained during the first 8 months of stimulation (weeks 1–32). After successful pain control from week 37, the Oxcarbazepine dose was reduced to 300 mg BID, and Pregabalin 75 mg BID was added. After pain relapse in weeks 77–78, Oxcarbazepine and Pregabalin doses were increased to 375 mg and 125 mg, respectively, in the morning and 450 mg and 200 mg, respectively, in the evening. A local lidocaine therapy overnight was also introduced. After that, pain control was maintained, and medication doses were further reduced. The Oxcarbazepine morning dose was reduced to 300 mg in week 82, 150 mg in week 90, and discontinued in week 91, while the evening dose was reduced to 300 mg in week 82, 150 mg in week 87, and discontinued in week 90. On the other hand, Pregabalin’s morning dose was reduced to 100 mg in week 84, 75 mg in week 87, and discontinued in week 92, while the evening dose was reduced to 100 mg in week 82, 75 mg, and 50 mg in week 85, 25 mg in week 86 and discontinued in week 87.

### 2.3. Repetitive Transcranial Magnetic Stimulation (rTMS)

The stimulation was performed using a MagPro X100 magnetic stimulator with an MCF B65 figure-of-eight coil (MagVenture A/S, Farum, Denmark). For precise coil placement during stimulation, rTMS was performed with Localite TMS Navigator software (LOCALITE Biomedical Visualization Systems GmbH, Sankt Augustin, Germany). The patient’s T1-weighted MRI scan was used for head and coil registration, target planning, and neuronavigation during stimulation. During rTMS sessions, the patient was seated in a comfortable chair with a head and armrest. Six cortical targets in the left precentral gyrus were defined, covering the contralateral hand and face area (Figure 2). On the hand area, single-pulse TMS was delivered to locate the right abductor pollicis (APB) muscle representation and determine the resting motor threshold (RMT). Electromyography (EMG) recordings from the right APB muscle were obtained using surface Ag-AgCl electrodes (9 mm diameter) with a belly-tendon montage. RMT was defined as the minimum TMS intensity that produced a motor-evoked potential (MEP) of about 50 µV amplitude in 50% of 10 trials at rest. The TMS coil was placed tangentially on the scalp, with the handle pointing laterally and caudally at a 45° angle from the mid-sagittal line, so the induced current flowed in a posterior–anterior direction. The patient’s RMT was measured every session before the rTMS protocol. During rTMS stimulation, 500 biphasic pulses (100 pulses per train) were delivered at 10 Hz for each target with 20 s inter-train-interval [15]. The stimulation intensity was set at 90% of the patient’s RMT. The magnetic coil was always held perpendicular to the cortical target. Each rTMS session typically lasted for approximately 30 min, including the preparations. The patient tolerated all stimulation sessions and reported no adverse effects (e.g., headache or dizziness). The stimulation protocols conformed to the safety guidelines for rTMS [16].

## 3. Analysis

The analyses were performed using SPSS version 27 software (IBM Corp., Armonk, NY, USA). The primary outcome variable was the numeric pain intensity scores derived using the visual analog scale (VAS). In a 10 cm line, with two endpoints representing 0 (“no pain”) and 10 (“the worst pain imaginable”), the patient is instructed to mark the point that they feel represents their pain intensity. The VAS score was measured three times a day (morning, afternoon, and evening) on four consecutive days one week before starting the stimulation (week 1: baseline phase) and then three times daily once the stimulation was started (weeks 2–89: intervention phase). The median weekly pain scores were calculated and plotted (Figure 3). Data patterns, trends, and overlaps were checked. Using an online calculator, tau-U was determined to detect data overlap between the intervention and baseline phase (http://www.singlecaseresearch.org/calculators/tau-u, accessed on 1 September 2022 [14]. Then, to indirectly quantify an overlap, if present, the percentage of non-overlapping data (PND) was calculated by hand (non-overlapping data points/all the data points × 100) [17]. The baseline and intervention phase individual median weekly pain scores were identified and statistically compared using the Wilcoxon matched-pairs signed rank test (two-tailed). The data after the balloon compression procedure were not presented and included in the analysis because, after that, it is difficult to disentangle this procedure’s influence and that of rTMS on pain ratings. Cohen’s *d* was calculated as an index of overall effect size. A *p*-value of <0.05 was considered significant for all statistical analyses.

## 4. Results

The patient completed 90 weeks of interventions. The patient reported 191 pain-free days during the first 80 weeks of treatment, and the average daily pain score during stimulation was 1.02. The baseline median pain score was 3.61 (week 1); therefore, scores during the intervention phase that are lower and higher than it could indicate relief and worsening of pain, respectively. Visual inspection of the data showed that pain scores during the intervention phase were mostly lower than the baseline score except on two occasions (week 36: 3.29; week 63: 3.29), where the pain intensity nearly reached the baseline level (Figure 2). The overall reduction in pain score was 70.89% (weeks 2–81 pooled score: 1.04). Excluding the data from weeks 36 and 63 decreased the pooled score to 0.98, indicating a 72.55% overall reduction in the pain score during the intervention phase. There were no monotonic trends in the baseline data (Tau-U = −0.167, *p* = 0.734); hence no correction was applied when contrasting the baseline (A) and intervention (B) phases. The result of the contrast: Tau-U = −0.988, Z = −3.319, *p* = < 0.001, indicates the absence of overlap between AB phases showing differences in the overall baseline and intervention pain scores. This was consistent with the PND of 97.5%, indicating a negligible overlap (2.5%) of the intervention phase data with the lowest data point in the baseline phase. The PND score suggests that rTMS effectively reduced the pain scores (<50 = no observed effect, 50–70 = questionable effect, and >70 = the intervention was effective) [17]. On the other hand, the Wilcoxon matched-pairs signed rank tests (Appendix A) indicate statistically significant reductions in weekly pain scores compared to the baseline pain score. The only exceptions to these findings were observed in week 2 (*Z* = −1.913, *p* = 0.056), week 36 (*Z* = −0.594, *p* = 0.553), and week 63 (*Z* = −0.314, *p* = 0.753). Cohen’s *d* was 0.523, indicating an overall medium effect size.

## 5. Discussion

In this patient with refractory TN, we explored the therapeutic potential of long-term rTMS stimulation. To this end, the patient received 73 high-frequency rTMS treatments over M1 contralateral to the affected face over 23 months. A trend-wise decrease in pain scores was observed during the intervention phase, and statistical analysis confirmed it as a significant reduction in pain intensity. The present result points to short-term and long-term relief of painful symptoms with repeated 10 Hz rTMS over M1 in TN and the safety of longitudinal multi-session stimulation in clinical settings.

The overall pain relief elicited in our patient (70.89%) was consistent with the 33.66% pain reduction observed in another single refractory TN patient after 35 10 Hz rTMS stimulation sessions over 1 year [12]. Pain was also well controlled in 24 patients (45%) after 5 daily rTMS sessions of the left M1 hand area [10], in 12 patients after a single-session stimulation (crossover design) of the left (22.92%) and bilateral (42.59%) M1 hand area [9], and in 12 patients after a single-session stimulation (crossover design) of the left M1 hand (13%) and face (5–6%) area [18]. Pain level was also reduced (58.3%) by a sham-controlled single session of 10 Hz rTMS to the left M1 hand area in 12 patients with trigeminal nerve lesions [19]. The greater pain reduction in our patient can be due to the higher number of treatment sessions (23 months) and shorter inter-session intervals (maximum 4 weeks in our case vs. 4 months in the Zaghi et al. study). In addition, contrary to the aforementioned studies that stimulated the M1 hand region, we delivered rTMS to both hand and face representations in the contralateral M1. It is challenging to delineate the respective contribution of M1 hand and face area stimulation on pain relief. It can be that the effect of 10 Hz rTMS spread to both sites modulating their respective output to the other area or that the face representation may have shifted towards the hand area, such as in cases of facial lesion and cortical reorganization induced by deafferentation due to chronic pain [19]. Nonetheless, we could argue that facial area stimulation contributes to overall pain relief. In support of this assumption, it was shown that somatotopic matching between pain topography and M1 stimulation site (e.g., facial pain and stimulation of the face representation in M1) is a predictive factor for the positive effect of rTMS and surgically implanted epidural M1 stimulation, particularly in the presence of trigeminal nerve lesions and pain location at the face [18,19,20].

In our patient, the effect of rTMS was not also immediate, as shown by the insignificant reduction of pain in week 2, where she had four daily stimulation sessions. This is consistent with the previous observation that the effect of repeated stimulation works cumulatively, needing at least two to four days to build up [10,19]. The pattern of our results also agrees with the proposal that significant pain relief could be limited to about two weeks after the end of stimulation [10,12]. This is because the patient experiences increased pain intensity in weeks 36 and 77–78, which are preceded by 3-week and 4-week intersession intervals, respectively. Pain intensity also increases in week 63, which follows a once-per-week stimulation session (weeks 37–62) with 1–2 weeks intersession intervals. For this, we could argue that the two-week effective pain control can also wane when the stimulation is delivered regularly for a long period, probably due to a ceiling effect.

The precise neurophysiological mechanism underlying pain relief with rTMS in TN and other craniofacial pain syndromes is still unclear. There is an idea that pain transmission strictly depends on the balance of the excitatory and inhibitory influences that act on the neuronal circuits of the somatosensory system [21]. In TN, this balance is thought to be compromised by central sensitization induced by trigeminal nerve injury on the nociceptive neurons, particularly the caudalis subnuclei (the “medullary dorsal horn”) involved in processing facial nociceptive signals [22]. Sensitization alters the input/output relationship of neurons in the trigeminal nociceptive pathways (trigeminal nerve → trigeminal brainstem sensory nuclear complex (VBSNC) → thalamus → primary (S1) and secondary somatosensory cortex and insula), making them hyperexcitable, resulting in amplified nociceptive and non-nociceptive signals [22,23,24]. There is growing evidence supporting the presence of aberrant and overactive central sensory transmission in TN. For instance, pain-related evoked potential amplitudes were larger and latencies shorter in TN patients with chronic facial pain than in those without facial pain [25]. Most recently, in familial trigeminal neuralgia, it was shown that expression of TRPM7 mutant channels results in an abnormal Na+ influx that depolarizes the membrane potential of trigeminal ganglion neurons, causing hyperexcitability [26].

There is an anatomical and functional reciprocal M1-S1 connection. Therefore, there is a possibility that an exaggerated ascending noxious signal, including those mediated by the ventrobasal complex and posterior thalamic nuclei-S1 route (the sensory discriminative component or pain) and the intralaminar thalamic and posterior aspect of ventromedial thalamic nuclei-S1 and limbic route (the affective-motivational component of pain) [27], would be affected by M1 stimulation. Theoretically, the rTMS-elicited increase in M1 neuronal excitability could reduce pain because it may dampen S1 processes, causing an interruption to the final step in noxious signal processing, the interpretation of sensory information. Another possibility is the direct activation of the perigenual cingulate and orbitofrontal areas by descending inputs from M1 that would influence the emotional appraisal of pain rather than its intensity [28]. M1 stimulation could also facilitate the top-down activation of brainstem periaqueductal gray, leading to descending inhibition toward the spinal cord [28]. Indeed, it was found that the effects of rTMS in M1 are more long-lasting for affective than for sensory pain [29] and that the analgesic effects could result, at least partly, from the restoration of defective intracortical inhibitory processes [30].

## 6. Limitations of the Study

The absence of sham stimulation poses a limitation to this report because variables unrelated to the stimulation may confound the assessment of pain intensity (e.g., favorable changes in the patient’s personal life that can affect her mood). Moreover, for safety reasons, the patient’s medications are not discontinued during the intervention phase. Therefore, the absence of sham stimulation also made it difficult to isolate the effect of rTMS on pain from the combined effect of rTMS and medications. Nevertheless, from a technical point of view, it would be challenging to perform sham stimulation because the patient can easily differentiate it from active rTMS stimulation. The M1 face area is very lateral, and rTMS can induce noticeable facial twitches even at a subthreshold intensity due to ipsilateral facial nerve stimulation [19]. In other words, facial responses induced by cortical activation are hard to disentangle from those induced by peripheral facial nerve stimulation [18]. In addition, qualitative reporting of all changes in the patient’s social, physical, and emotional state must be documented. This is important because VAS is one-dimensional, as it only measures pain intensity and therefore provides incomplete treatment outcome assessments. Other dimensions disregarded in VAS include physical functioning, emotional functioning, participant ratings of improvement and satisfaction with treatment, symptoms and adverse events, and participant adherence to treatment regimens [31]. A further limitation is the single-patient experimental design. Although this allows detailed analysis of within-subject variability, the generalization of the results must be cautioned. Studies with a bigger sample size must test the efficacy of the stimulation paradigm employed in this single-case study in the future.

## 7. Conclusions

This single-case study showed that 10 Hz rTMS could be an effective adjunct treatment method for patients with refractory TN. As shown in our patient, long-term repeated application of 10 Hz rTMS to M1 is not associated with serious complications. These results provide encouraging insights for designing patient-tailored brain stimulation therapies to induce long-lasting pain relief. Nonetheless, our preliminary results underline the need for more patient studies to investigate the therapeutic effects of long-term rTMS application in TN.

## Figures and Tables

**Figure 1 life-13-00126-f001:**
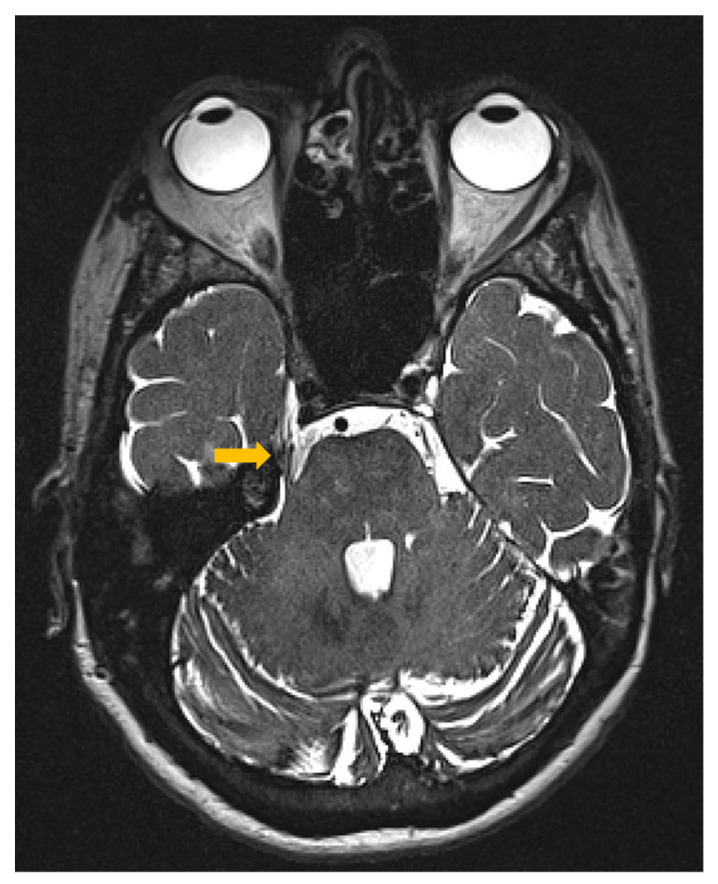
The patient’s brain axial MRI image shows the right superior cerebellar artery contacting (yellow arrow) the superior aspect of the trigeminal nerve route without causing marked displacement. MRI = magnetic resonance imaging.

**Figure 2 life-13-00126-f002:**
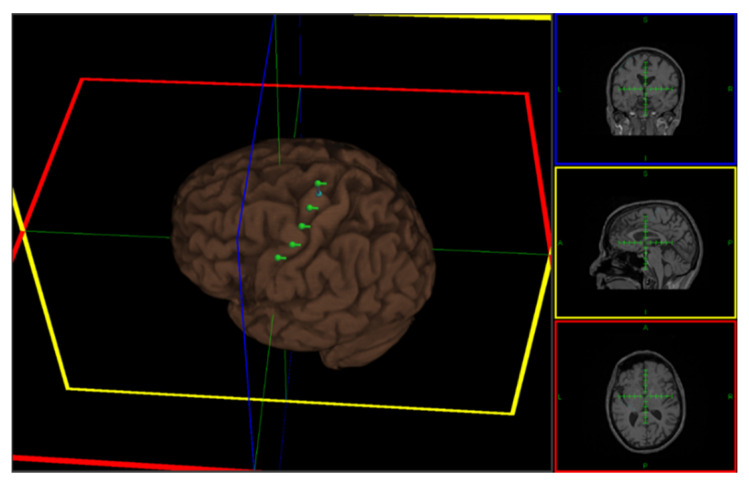
rTMS stimulation targets in the left precentral gyrus. The target areas (green landmarks) include the representation of the right abductor pollicis muscle (hand) and affected face area separated by a 1cm inter-target distance. Colored lines mark the center of the image shown on the right side.

**Figure 3 life-13-00126-f003:**
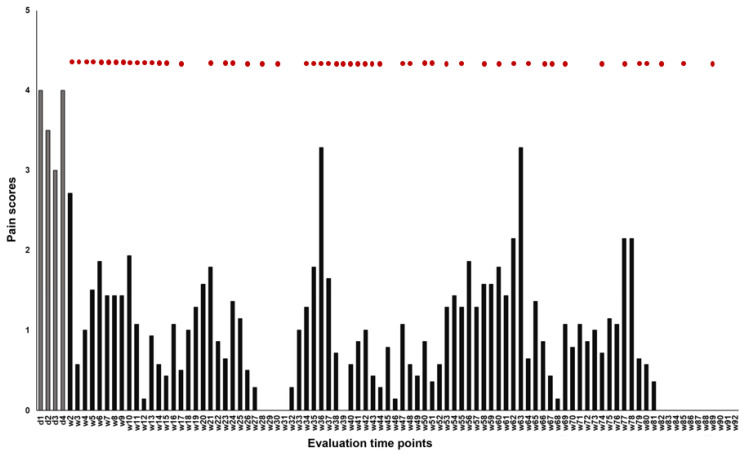
Effect of rTMS on subjective pain scores. The y-axis depicts the median pain intensity assessed by a 10-point visual analog scale (VAS), 0 (“no pain”) and 10 (“the worst pain imaginable”). The x-axis depicts the time points of measurements. Gray bars represent daily pain scores in the baseline phase (week 1), and black bars represent weekly median pain scores in the intervention phase (weeks 2–92). Red dots indicate weeks when the patient received rTMS stimulation. d = day, w = week.

## Data Availability

All obtained data is available in the case report or in the Appendix A.

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
