# Peer review of "Twenty-Three Months Repetitive Transcranial Magnetic Stimulation of the Primary Motor Cortex for Refractory Trigeminal Neuralgia: A Single-Case Study"

_life, 2023, doi:10.3390/life13010126_

Round 1
Reviewer 1 Report
I carefully reviewed the case report titled: “Twenty-three months repetitive transcranial magnetic stimulation of the primary motor cortex for refractory trigeminal neuralgia: a single-case study”, however, minor and major modifications are needed as follows:
Major
Introduction line 28, TN is usually unilateral however, could be bilateral as well, so modify
Line 32, according to the HIS Classification ICHD-3 the correct term is Classical trigeminal neuralgia, not primary TN.
Line 64 an only single case report cannot test the efficacy of any intervention, so I suggest changes the aim to test the long-term effect
Experimental assessment and therapeutic intervention section
Is unclear what mean AB or phase change without reversal design.
I recommend including a graph of this section with X and Y axes showing the data
In figure 2 I didn’t identify the VAS week 1 baseline phase.
In patient medical history is unclear when the TN started and the daily doses of the medications line 72. The name of the disease is incomplete for hypertension and is undefined the type of depression of the patient and if refractory as well.
Minor
Grammatical errors in the abstract as “consenting” line 15 or introduction line 30 suffices,
Author Response
We thank the editor and the reviewers for allowing us to revise our paper "Twenty-three months repetitive transcranial magnetic stimulation of the primary motor cortex for refractory trigeminal neuralgia: a single-case study". Your insightful comments and suggestions are appreciated. We considered all of them in the revised version of our manuscript. We hope that the revisions made clarify the raised issues and help to improve the quality of the manuscript. We have included the reviewer's comments below and responded to them individually, describing the changes we have made in the manuscript. Revisions are marked as tracked changes in red in the revised manuscript. All authors have approved the revisions.

Reviewer 2 Report
Authors have proven the effectiveness of twenty-three months of repetitive transcranial magnetic stimulation of the primary motor cortex for refractory trigeminal neuralgia in a single case study. Their observations confirmed the efficacy and safety of the of 10 Hz rTMS therapy. The therapy allowed for the reduction of the medication dosages. Simulations were applied to the left motor cortex's hand and face representation areas precisely with the neuronavigation. In fact, authors confirmed revelations described in refs. Hanna J, Shapiro P, Gover-Chamlou A, Komarczyk E. Case Report: Repetitive Transcranial Magnetic Stimulation for Comorbid Treatment-Resistant Depression and Trigeminal Neuralgia. J ECT 2019;35:e37–e38.; Henssen DJHA, Hoefsloot W, Groenen PSM, Van Cappellen van Walsum AM, Kurt E, Kozicz T, van Dongen R, Schutter DJLG, Bartels RHMA. Bilateral vs. unilateral repetitive transcranial magnetic stimulation to treat neuropathic orofacial pain: A pilot study. Brain Stimul 2019;12:803–805.; Khedr EM. Longlasting antalgic effects of daily sessions of repetitive transcranial magnetic stimulation in central and peripheral neuropathic pain. J Neurol Neurosurg Psychiatry 2005;76:833–838.
The state of art in the Introduction section is presented clearly, as well as the aim. Some sentences require stylistic re-writing e.g. line 61 62 …”In the present case study, we report the case…”
Materials and Methods section includes all necessary data to repeat the study including a clear explanation of the applied rTMS algorithm. Did the authors forget about the exclusion criteria?
Results are presented well, applied statistics have been selected correctly, including indications by Lenz AS. Calculating Effect Size in Single-Case Research. Meas Eval Couns Dev 2013;46:64–73. and Lobo MA, Moeyaert M, Baraldi Cunha A, Babik I. Single-Case Design, Analysis, and Quality Assessment for Intervention Research. J Neurol Phys Ther 2017;41:187–197.
In the Discussion section, the explanation of Authors on …”as it appeared that somatotopic matching between pain topography and M1 stimulation site, particularly in the presence of trigeminal nerve lesions and pain location at the face”… , supported by refs. [2,12,17], is puzzling and needs re-writing. Apart of this, the neurophysiological explanation of rTMS in TN is interesting (lines 244-276). Study limitations are satisfactory.
Please mention …”10Hz rTMS”… in the Conclusions.
References are selected correctly, but their editing (style of citation) is not in line with MDPI's expectations. The paper needs the editorial corrections of refs. applying throughout the text.
Author Response

(The authors gave the same response as above.)

Reviewer 3 Report
Thank you for permitting me to review this manuscript
This is a one case experimental treatement without control or sham stimulation in onlyone patient , it is difficult to rule out a a natural evolution or a placebo effect however the manuscript is well written
medical treatement side effects also include sedation which should be included in the list
If possible please provide initial MRI of the compression before vascular decompression
With regards the pain scores it is difficult to undermine natural evolution of pain the authors should provide more information to readers as this was not natural course of the disease in this special patient
It appears also that medications was not stopped so it is really difficult to relate duration of pain score reduction only to the stimulation case
Please provide a reference for the protocol of stimulation
reference 20 appears to be a safety protocol , is it also available for treatement ?
Table 1 is un necessary and can be resumed in a very short text
Author Response

(The authors gave the same response as above.)

Round 2
Reviewer 1 Report
None
Reviewer 3 Report
The author has improved the manuscript as a very well written and structured case report